# Pituitary Hormones (FSH, LH, PRL, and GH) Differentially Regulate AQP5 Expression in Porcine Ovarian Follicular Cells

**DOI:** 10.3390/ijms20194914

**Published:** 2019-10-03

**Authors:** Mariusz T. Skowronski, Patrycja Mlotkowska, Damian Tanski, Ewa Lepiarczyk, Bartosz Kempisty, Lukasz Jaskiewicz, Chandra S. Pareek, Agnieszka Skowronska

**Affiliations:** 1Veterinary Center, Nicolaus Copernicus University in Torun, 87-100 Torun, Poland; chandra.s.pareek@gmail.com; 2Department of Animal Physiology, The Kielanowski Institute of Animal Physiology and Nutrition, Polish Academy of Sciences, 05-110 Jabłonna, Poland; patrycja.mlotkowska@wp.pl; 3Department of Animal Anatomy and Physiology, University of Warmia and Mazury in Olsztyn, 10-719 Olsztyn, Poland; damian.tanski@uwm.edu.pl; 4Department of Human Physiology, School of Medicine, University of Warmia and Mazury in Olsztyn, 10-752 Olsztyn, Poland; ewa.lepiarczyk@uwm.edu.pl (E.L.); lukasz.jaskiewicz@uwm.edu.pl (L.J.); 5Department of Histology and Embryology; Poznan University of Medical Sciences, 61-701 Poznan, Poland; bkempisty@ump.edu.pl; 6Department of Anatomy, Poznan University of Medical Sciences, 61-701 Poznan, Poland; 7Department of Obstetrics and Gynecology, University Hospital and Masaryk University, 602 00 Brno, Czech Republic; 8Interdisciplinary Center of Modern Technology, Nicolaus Copernicus University in Torun, 87-100 Torun, Poland

**Keywords:** Aquaporin 5, gene and protein expression, ovarian cells, in vitro study, pig

## Abstract

This study aimed to examine the effect of follicle-stimulating hormone (FSH), luteinizing hormone (LH), prolactin (PRL), and growth hormone (GH) on Aquaporin 5 (AQP5) expression in granulosa (Gc) and theca cells (Tc) from medium (MF) and large (LF) ovarian follicles of pigs. The results showed that GH significantly decreased the expression of AQP5 in Gc from MF in relation to the control. In the Gc of large follicles, PRL stimulated the expression of AQP5. However, the increased expression of AQP5 in the Tc of LF was indicated by GH and PRL in relation to the control. A significantly higher expression of the AQP5 protein in the Gc from MF and LF was indicated by FSH and PRL. In co-cultures, an increased expression of AQP5 was observed in the Gc from LF incubated with LH, PRL, and GH. A significantly increased expression of AQP5 was also observed in co-cultures of Tc from all type of follicles incubated with LH, whereas PRL stimulated the expression of AQP5 in Tc from MF. Moreover, AQP5 protein expression increased in the co-culture isolated from MF and LF after treatment with FSH, LH, PRL, and GH. AQP5 immunoreactivity was observed in the cytoplasm, mainly in the perinuclear region and endosomes, as well as in the cell membranes of Gc and Tc from the LF and MF.

## 1. Introduction

Fluid transport across cellular barriers in biological tissues results from water transport driven by osmotic gradients or hydrostatic pressure differences. The aquaporins (AQPs) are a family of small hydrophobic intra-membranous proteins (25–34 kDa), functioning as cell membrane water channels facilitating water and small neutral solute transport across a variety of biological membranes. Transcellular water flow is dependent on the permeability of the plasma membrane to water molecules [1,2].

AQPs are expressed in a wide range of tissues, often spatially located within a certain region of the cell. This enables them to play a central role in physiological processes in the flow of water through those tissues, which typically triggers the cell volume regulation mechanism. The involvement of these small proteins in cell adhesion, migration, proliferation, and cell differentiation has also been defined [3]. The important role of AQPs was discovered following the observation of a mouse model with silenced genes (*Aqp-null mice*), e.g., the involvement of AQPs in cell migration (AQP1) [4], fat metabolism (AQP7) [5] and glucose metabolism (AQP9) [6]. The expression of aquaporins varies, depending on the tissue. Furthermore, any one cell may possess several isoforms, often located at opposite poles of the cell, in the apical membranes and basolateral membranes. There is now an emerging consensus that the rapid and reversible translocation of AQPs from intracellular vesicles to the plasma membrane is trigged by a range of stimuli. For example, as a result of stimulation of the α_1_-adrenergic receptor, AQP5 translocates from intracellular membranes to the apical membranes of rat parotid gland cells [7].

Twelve AQP isoforms are expressed in the mammalian reproductive system and play very important roles in maintaining water homeostasis in reproductive cells, and their dysfunction can lead to impaired fertility. Studies on the human uterus have demonstrated the mRNA expression of *AQP1, 2, 3, 4, 5, 7, 8, 9,* and *11* in chorion, amnion, and the placenta, but not *AQP2* [8,9]. Very recently, Klein et al. [10] demonstrated the uterine mRNA expression of 12 different AQP subtypes, while Western blot analysis confirmed the protein expression of AQP0, 2, and 5. The presence of AQPs was found in plasma membranes of all animal cells, but the mechanism of their action and the transcriptional level (in the mRNA and protein) are unclear. Transport and water homeostasis in the ovary, oviduct, uterus, placenta, and fetal membranes is crucial for the maintenance of a normal reproductive function, oocyte development, embryo implantation, and fetal growth. Depending on the physiological status of the reproductive system (phase of the cycle, implantation, placentation, and labor), there is a periodic agitation or silencing of many genes in their structures. The following question then arises: what is the role of hormones in the regulation and expression of AQP genes and proteins?

The first publications concerning the role of steroid hormones (estradiol, E_2_, and progesterone, P_4_) in the regulation of AQP1 and 5 expression in the uterine tissue of mice and rats showed their different effects. A direct action of E_2_ on AQP5 expression in a mouse uterus was found [11]. Then, other authors demonstrated that the administration of P_4_ and/or E_2_ regulates AQP1 and AQP5 expression in the uterus of ovariectomized rats [12]. However, in ovariectomized mice, the same treatment did not reveal similar changes. It was not until the earlier administration of P_4_, and then the administration of E_2_, that AQP5 expression was induced [13]. It was also demonstrated that after E_2_ treatment, *AQP1* mRNA expression increased in mice myometrium, but not the protein level. In turn, by applying immunocytochemistry and immunoprecipitation techniques, other researchers demonstrated a significant effect of estrogens on the localization of AQP1 in mice myometrium, without any effect on AQP5 expression [14]. Furthermore, the expression of human endometrial AQP2 occurs in the menstrual cycle because of the estrogen response element in the promoter region of the *AQP2* gene. AQP2 is involved in water transport in the human endometrium and might play a role in the cyclic changes of the endometrium, suggesting a role for AQP2 in implantation, edema, and/or menstruation [15,16]. Ducza and others [17] demonstrated the effect of oxytocin on AQP5 expression in a rat uterus during pregnancy. Recently, this research group indicated an influence of female sexual hormones on the expression of AQP5 in a late-pregnant rat uterus [18,19].

Reports on AQP expression in the ovary are not as numerous as in the uterus. Experiments have shown that AQPs are involved in the process of pig folliculogenesis [20], as well as in the creation of the vesicle cavity (antrum) in rodents and humans [21,22]. McConnell et al. [23] showed that the rapid flow of water to rat follicle cavities mainly occurs via the transmembrane pathway in granulosa cells (Gc), probably involving AQP7, AQP8, and AQP9. Recently, studies by Starowicz et al. [24] showed the presence of AQP5 in preovulatory ovarian follicles and its role in the process of cumulus oophorus expansion. Rodges and Irving-Rodgers’ report indicated that the production of hyaluronan by Gc generates an osmotic gradient that causes fluid to move from the theca cells (Tc) to the Gc, facilitating the transcellular flow of water [25].

Inspiration for this study came from the earlier results obtained from pigs. These results suggest that functional and distinctive collaboration exists among diverse AQPs in a pig’s ovary, oviduct, and uterus during the estrous cycle and early pregnancy [19,26,27]. We observed the specific pattern of expression and established an anatomical basis for examined AQPs, which indicates that they play important functions in the regulation of water movement to maintain adequate hydration and the regeneration ability of the reproductive tract. Recently, our study found that the follicle-stimulating hormone (FSH), luteinizing hormone (LH), prolactin (PRL), and growth hormone (GH) regulated AQP1 expression in the ovarian follicular cells of pigs [28].

Hence, the aim of this study was to establish whether the above mentioned biological factors also participate in the regulation of mRNA and protein expression of AQP5 in porcine follicular cells.

## 2. Results

### 2.1. The Effects of FSH, LH, PRL, and GH on AQP5 mRNA Expression in Gc and Tc Cells from the Medium (MF) and Large (LF) Follicles

AQP5 mRNA expression significantly decreased in the Gc cells isolated from medium follicles after a 24 h culture with GH compared to the control (*p* < 0.05; Figure 1A). PRL increased AQP5 mRNA expression in the Gc cells obtained from large follicles (*p* < 0.05; Figure 1B). In the Tc obtained from large follicles, PRL and GH significantly increased AQP5 mRNA expression (*p* < 0.05; Figure 1D). Other treatments did not affect AQP5 mRNA expression in both cells at the examined time point in comparison to the control group (Figure 1A–D).

### 2.2. The Effects of FSH, LH, PRL, and GH on AQP5 Protein Expression in the Gc and Tc Cells from MF and LF

SDS-PAGE and Western blot analysis revealed that FSH and PRL increased the AQP5 protein expression in porcine GC cultured for 24 h obtained from medium and large follicles (*p* < 0.05; Figure 2A,B) in comparison to the control. AQP5 protein was detectable in Tc isolated from medium and large porcine follicles, but the tested factors did not affect the AQP5 expression.

### 2.3. The Effects of FSH, LH, PRL, and GH on AQP5 mRNA Expression in the Co-Culture of Gc and Tc Cells from MF and LF

The co-culturing of Gc with theca cells from the large follicles in the presence of LH, PRL, and GH significantly increased AQP5 mRNA expression compared to the respective control (*p* < 0.05; Figure 3B). The addition of LH and PRL produced a significant increase in AQP5 mRNA expression in the Tc coculture with granulosa cells of medium follicles (*p* < 0.05; Figure 3C). A stimulatory effect on AQP5 mRNA was observed in the Tc in the presence of LH (*p* < 0.05; Figure 3D). Other treatments did not affect AQP5 mRNA expression in both cells at the examined time point in comparison to the control group (Figure 3A–D).

### 2.4. The Effects of FSH, LH, PRL, and GH on AQP5 Protein Expression in the Co-Culture of Gc and Tc Cells from MF and LF

AQP5 protein expression significantly increased in the co-culture of Gc with Tc isolated from medium and large follicles after treatment with FSH, LH, PRL, and GH, respectively (*p* < 0.05; Figure 4A,B). In the co-culture of Tc with Gc obtained from medium follicles, AQP5 protein expression significantly increased with FSH, LH, PRL, and GH (*p* < 0.05; Figure 4C). In large follicles, this effect was only observed in the presence of GH (*p* < 0.05; Figure 4D).

### 2.5. The Effects of FSH, LH, PRL, and GH on AQP5 Protein Expression in the Culture of Gc and Tc Cells from MF and LF

Immunofluorescence found that AQP5 protein was localized in the cytoplasm, mainly in the perinuclear region and endosomes, and weak labeling was observed in the cell membranes of Gc (Figure 5A–D) and Tc (Figure 6A–D) cells from the LF, as well as Gc (Figure 5E–H) and Tc (Figure 6E–H) cells from the MF. There was no effect on the distribution in these cells of AQP5 protein after 24 h treatment with FSH, LH, PRL, and GH compared to the controls (Figure 5 and Figure 6).

## 3. Discussion

Inspiration for this study came from the earlier results obtained from the pig [28]. The study found that follicle-stimulating hormone (FSH), luteinizing hormone (LH), prolactin (PRL), and growth hormone (GH) regulated AQP1 expression in the ovarian follicular cells of pigs. The present results suggest that AQP5 can also be regulated by FSH, LH, PRL, and GH.

In previous research [28] we found a stimulatory effect of FSH and LH on AQP1 mRNA and protein expression in the Gc ant Tc of pig medium and large ovarian follicles and co-cultures of these cells. In the present research, AQP5 was mainly submitted to the regulatory effect of FSH and LH, since these hormones markedly increased AQP5 mRNA and protein expression in Gc and Tc, as well as in co-cultures of the cells. Therefore, it may be assumed that FSH and LH exert their effect on porcine follicles by influencing the mRNA and protein expression of AQP1 and AQP5.

Previous studies reported an effect of steroid hormones on AQP expression in the female reproductive system [14,23]. Estrogens up-regulated *AQP2* and *AQP5* by the promoter regions, as shown in the human and rodent uterus [11,16]. These hormones stimulate follicular cell proliferation and, together with FSH, initiate the formation of LH receptors in granulosa cells, as well as progesterone and androgen production in theca cells [29,30]. Our previous papers presented the influence of steroid hormones on AQPs in the female reproductive system [31,32]. Others have presented very interesting results on transgenic mice deficient in AQP proteins due to different reproductive phenomena [33,34,35,36]. Our previous studies [37] demonstrated AQP5 and AQP9 in porcine granulosa cells and AQP1 in theca cells. Zhu et al. [38] found the mRNA expression of AQP1, 3, 4, 5, 6, 7, 8, 9, and 11 in a pig placenta on day 25 of gestation. The authors also identified AQP1, 3, 5, and 9 protein expression in the placenta, uterine endometrium, and porcine trophectoderm cell lines.

Ovarian follicle growth, differentiation, and steroidogenic activity are controlled by many factors, of which FSH, LH, PRL, and oxytocin are the most important [39]. Moreover, the authors have emphasized that the growth of follicles and steroidogenesis is controlled by the interaction between insulin-like growth factors (IGF-S) and gonadotrophins. Attention has also been focused on GH, which affects the follicular function by the hypothalamic-pituitary axis or acting directly on the ovary. The direct effects of GH on steroidogenesis in porcine granulosa cells have been found [40]. Moreover, Kolodziejczyk et al. [41] found that Gc and Tc produce IGF-I and showed the effect on proliferation of the cells. Growth hormone, LH, and sex steroids are necessary for proper ovarian angiogenesis due to their influence on local factors, mainly the vascular endothelial growth factor A (VEGF-A) [42]. Research involving a pRNA-H1.1 vector containing the short hairpin RNA (shRNA) targeting *AQP5* mRNA expression in human umbilical vein endothelial cells (HUVECs) found that the *AQP5*-silenced HUVECs acquired decreased proliferation, migration, and the tube formation ability. Furthermore, the expression and secretion of VEGF-A in colorectal cancer cells were downregulated by *AQP5* shRNA [43].

We recently demonstrated a stimulatory effect of PRL on AQP1 protein expression in the co-cultures of Gc and Ts of medium and large follicles [28]. GH increased the expression of AQP1 protein in Gc isolated from large follicles. Furthermore, the expression of *AQP1* mRNA was significantly higher under the influence of GH in co-cultures of Gc of large follicles, as well as Tc of medium follicles. The present study found that *AQP5* mRNA expression decreased in the Gc from medium follicles with GH. PRL increased *AQP5* mRNA expression in the Gc in large follicles, as well as PRL and GH did in the Tc. Protein analysis revealed that PRL increased AQP5 expression in Gc obtained from medium and large follicles. The co-culturing of the Gc and Tc cells from the large follicles in the presence of PRL and GH increased *AQP5* mRNA expression. The addition of PRL produced an increase in *AQP5* mRNA expression in the co-culture of cells of medium follicles. AQP5 protein expression significantly increased in the co-culture of cells isolated from medium and large follicles after treatment with PRL and GH. In the co-culture of these cells obtained from medium follicles, AQP5 protein expression significantly increased with PRL and GH in turn, in large follicles, and this effect was only observed in the presence of GH. Other research found the influence of osmoregulatory hormones such PRL, GH, and cortisol on AQP3 expression in the gill epithelium of *Mozambique tilapia* [44].

Steroid and protein hormones are produced by follicles, which, by auto- and paracrine paths, affect ovarian follicular cell functions. In the present experiment, disparity between the level of mRNA transcripts and their corresponding protein was observed. Studies frequently report that cellular protein concentrations are not strongly correlated with the abundance of their corresponding mRNAs. We have observed this disparity in our study for the tested AQPs. Recent reports indicate that only 40% of the variation in protein levels can be explained by mRNA, and this may also result from the differentiated stability of mRNAs/and or proteins. It is also noteworthy that the processes of transcription and translation are not equally efficient [45]. A significant role in these processes is dependent on miRNA, and their sequences might be complementary to one or several molecules of mRNA. Recent estimates suggest that between 30% and 50% of the genes may be regulated by miRNAs.

Our recent study on a swelling assay using an AQP blocker found the presence of AQPs in Gc and Tc [28]. The increase in volume of Gc of medium and large follicles in hypotonic conditions after treatment with FSH, PRL, and GH was higher compared to the Tc of medium and large follicles under the same osmotic conditions with LH, PRL, and GH. It is likely that Gc have more AQP isoforms in the cell membranes than the Tc, which is in agreement with the present data on separated granulosa and theca cells (Figure 2).

AQP5 expression was also confirmed by immunofluorescence analysis. AQP5 was mainly localized in the perinuclear part of the cytoplasm and endosomes, and weak labeling was observed in the cell membranes of the pig Gc and Tc from MF and LF. In previous experiments, we found a similar subcellular distribution of AQP1 to AQP5 in these cells [28]. Several studies have shown that there is a decoupling of AQP1 and AQP5, suggesting that other factors need to be considered to study the hormones that regulate one or the other aquaporins. This is seen in conditions like stimulation by P_4_, E_2_, arachidonic acid (AA), oxytocin, cyclic adenosine monophosphate (cAMP), or forskolin (FSK) in pig uterus cells during early pregnancy [31,32]. Under the control condition, AQP1 was associated with the plasma membranes of endothelial cells, but AQP5 was mainly localized in the apical membranes of uterine epithelial cells. However, after treatment with P_4_, E_2_, AA, cAMP, or FSK, AQP1 localization did not change, but AQP5 labeling additionally appeared in the basolateral membranes of epithelial cells. In other experiments, the immunofluorescence of AQP7, AQP8, and AQP9 staining showed wide localization in cytotopes of a dog’s normal testis in either the tubular or interstitial partitions. In contrast, in the cryptic testis, these AQPs were only distributed in the interstitial partition of Leydig cells [46]. Recently, AQP5 was described as participating in protein–protein interactions, controlling the AQP2 isoform in kidneys [47]. There is much evidence suggesting the importance of AQP5 upregulation in tumor cell proliferation [48]. Moreover, Madeira et al. [49] have also demonstrated AQP5 as a factor implicated in adipocyte differentiation.

Based on our previous and present experiments, it might be concluded that AQP1 and AQP5 play important roles in maintaining water homeostasis in the ovarian cells of the pig. There is limited research on AQP expression in the reproductive tract of the pig, so the present research has provided some novel insights into the regulation of AQP5 by gonadotrophins, PRL, and GH in the Gc and Tc of porcine ovarian follicles. Folliculogenesis, ovulation, and corpus luteum formation and maintenance are processes that are critically dependent on appropriated water homeostasis, which may be established by the regulation of AQP5 expression by these hormones.

## 4. Materials and Methods

All experiments were performed in accordance with the Animal Ethics Committee (AEC approval No. 66/2010 DTN, 15 June 2010, University of Warmia and Mazury in Olsztyn, Poland). Tissue samples were recovered from mature cross-bred gilts (Large White × Polish Landrace) aged 7–8 months (*n* = 20), with an average weight of 90–110 kg, in a local slaughterhouse (Biskupiec, Poland). Ovaries were separated from all of the gilts and were then stored on ice and transported in cold-buffered physiological saline (PBS) supplemented with gentamycin and nystatin. The morphology of the ovaries has previously been evaluated [50]. The follicles were divided into two groups based on size: medium follicles (6–8 mm diameter) and large, pre-ovulatory follicles (9–12 mm).

### 4.1. Cell Cultures and Experimental Design

Granulosa and Tc were subsequently prepared according to the technique previously described [51] using a modification [52]. All stages of experiments were performed in sterile conditions. The total number of follicles (*n* = 10–12) for each group was used to obtain separated Gc and Tc. Using a pair of fine forceps, theca interna/granulosa were separated from external layers of the follicular wall. Gc were scrubbed from the follicular wall with round-tipped ophthalmologic tweezers and rinsed off by intensive pipetting (10 s) and a supernatant containing Gc was decanted. After isolation, Gc were rinsed in M199 medium with 5% bovine serum albumin (BSA) and centrifuged (180 × *g*, 10 min, 20 °C). Then, the cell pellet was treated with a red blood cell lysing buffer, as described by [28], and the Gc were re-suspended in M199 supplemented with 5% BSA and antibiotics and counted in Burker’s chamber. Cell viability was determined by trypan blue dye exclusion and was always greater than 90%. The Tc were mechanically separated from the underlying theca externa cell layer. Tc were washed with PBS, and exposed to trypsinization with 6–7 mL, 0.25% trypsin in PBS for 10 min at 37 °C. The cells were filtered through a nylon mesh. Finally, the cells were centrifuged and re-suspended in M199/BSA and antibiotics.

Experiment 1 was conducted to determine the effects of FSH, LH, PRL, and GH on the AQP5 mRNA expression in Gc and Tc cells. Incubation medium was M199 medium (Sigma, St. Louis, MO, USA) containing nystatin (120 U/mL) (Sigma) and gentamicin (0.05 mg/mL) (Krka, Novo Mesto, Slovenia). Aliquots of Gc were initially cultured in 1/12-well plates (Sarstedt, Equimed, Nümbrecht, Germany) with 1.0 × 10^6^ cells per well and Tc with 2.5 × 10^5^ cells per well without test compounds for 48–72 h, to allow cell attachment to 11 of the 16 wells (37 °C, 2% BSA, 10% fetal calf serum (FCS), 95% air/5% CO_2_) [30]. Following 48–72 h of attachment, cells were cultured with treatments (37 °C, 2% BSA, 5% FCS, 95% air/5% CO_2_) for the next 24 h, and 1.0 mL fresh M199/FCS alone was added to the control cultures, while FSH, LH, PRL, and GH were added to the experimental cultures at a concentration of 100 ng/mL (Sigma). When the experiments were terminated, the adherent cells were washed with PBS and then harvested and stored (−80 °C) for mRNA expression analysis.

Experiment 2 was conducted to determine the effects of FSH, LH, PRL, and GH on the AQP5 protein expression in Gc and Tc cells. Cultivations were conducted according to the procedure described above. Each treatment was conducted in four wells and each experiment was repeated three times. When the experiments were terminated, the cells (−80 °C) were collected and stored until all assays were completed for protein expression.

Experiment 3 was conducted for co-culture experiments to demonstrate the effects of FSH, LH, PRL, and GH on the AQP5 mRNA expression in Gc and Tc cells. For co-culture experiments, viable Gc and Tc were inoculated at a concentration of 2 × 10^6^ and 0.5 × 10^6^ cells/well, respectively, in tissue culture plates, which reflects typical ratios, as observed in vivo and described previously [31]. The media and incubation conditions with experimental factors were as described above. When the experiments were terminated, the cells (−80 °C) were collected and stored until all assays were completed for mRNA expression.

Experiment 4 was conducted for co-culture experiments to demonstrate the effects of FSH, LH, PRL, and GH on the AQP5 protein expression in Gc and Tc cells. For co-culture experiments, viable Gc and Tc were inoculated at a concentration of 2 × 10^6^ and 0.5 × 10^6^ cells/well, respectively, in tissue culture plates, which reflects typical ratios, as observed in vivo and described previously [28]. The media and incubation conditions with experimental factors were as described above. When the experiments were terminated, the cells (−80 °C) were collected and stored until all assays were completed for protein expression.

Experiment 5 was conducted to demonstrate the subcellular distribution of AQP5 protein in the Gc and Tc cells. Cells were isolated and cultured on Mini Cell slides (Merck Millipore, Burlington, MA, USA). Aliquots of Gc and Tc were 1.0 × 10^5^ cells per well/500 µL medium. The cells were cultured with treatments as described above (M199, 2% BSA, 5% FCS, 95% air/5% CO_2_) for the next 24 h. When the experiments were terminated, the cells were prepared for immunofluorescence.

### 4.2. RNA Extraction and Real-Time qPCR

Total RNA was isolated from Gc and Tc with the Total RNA kit (A&A Biotechnology, Gdynia, Poland), following the manufacturer’s recommendations, and quantified spectrophotometrically. The integrity of the product was confirmed on 1.5% agarose gel. Reverse transcription (RT) was performed using an Enhanced Avian HS RT-PCR Kit (Sigma Aldrich, St. Louis, MO, USA), and a mix of deoxynucleotides (dNTPs), and random hexamers as primers. The RT product was kept frozen at −20 °C for PCR analysis. Quantitative Real-Time PCR was used to establish dynamic changes in AQP5 mRNA expression. The following primer sequences were used [28,32]: AQP5 forward CTATGAGTCCGAGGAGGATT, AQP5 reverse GCTTCGCTGTCATCTGTT (NM_001110424.1), *GAPDH* forward GACCTCCACTACATGGTCTA, *GAPDH* reverse AAGATGGTGATGGCCTTTC (access No.: NM_001206359.1), *PPIA* forward GCACTGGTGGCAAGTCCAT, and *PPIA* reverse AGGACCCGTATGCTTCAGGA (access No.: AY266299), available in GeneBank. Glyceraldehyde 3-phosphate dehydrogenase (*GAPDH*) and Cyclophylin A (*PPIA*) were used as normalization controls. Real-Time PCR was performed (7300 Real-Time PCR system; Applied Biosystems, Foster City, CA, USA) as described previously [32]. Each experiment was independently repeated at least three times and the fold change in the expression of each gene was analyzed via the 2^−∆∆*C*t^ method.

### 4.3. SDS-PAGE and Western Blot Analysis

Gc and Tc were harvested, rinsed twice with PBS, lysed in radioimmunoprecipitation assay buffer (RIPA buffer) with protease inhibitors on ice for 30 min, and then centrifuged (12,000 × *g*) for 15 min at 4 °C. The protein concentration was determined by the Bradford method. Western blot analysis was performed as described previously by Skowronska et al. [32].

### 4.4. Immunofluorescence

Immunofluorescence was performed as previously described [28]. Briefly, cultures were fixed in 4% paraformaldehyde, rinsed with PBS, permeabilized with 0.2% saponin 0.01 M PBS for 10 min, and incubated for 30 min at 37 °C in PBS containing 10% normal goat serum (NGS). The slides were then incubated overnight at 4 °C with an anti-AQP5 antibody (1:200). After washing, coverslips were incubated with Alexa Fluor 488 donkey anti-rabbit IgG conjugated secondary antibodies for one hour. Nuclei were stained with TO-PRO^®^-3 (Invitrogen, Carlsbad, CA, USA). AQP5 localization was detected by fluorescent microscopy (Olympus, Tokyo, Japan).

### 4.5. Statistical Analysis

The data were analyzed by Statistica software (StatSoft Inc., Tulsa, OK, USA). The effect of the treatment was performed by a one-way analysis of variance for repeated measurements, followed by the LSD Fisher post-hoc test. Statistical significances were assigned at *p* < 0.05, while non-significant differences indicate *p* > 0.05. The data are presented as means ± S.E.M.

## Figures and Tables

**Figure 1 ijms-20-04914-f001:**
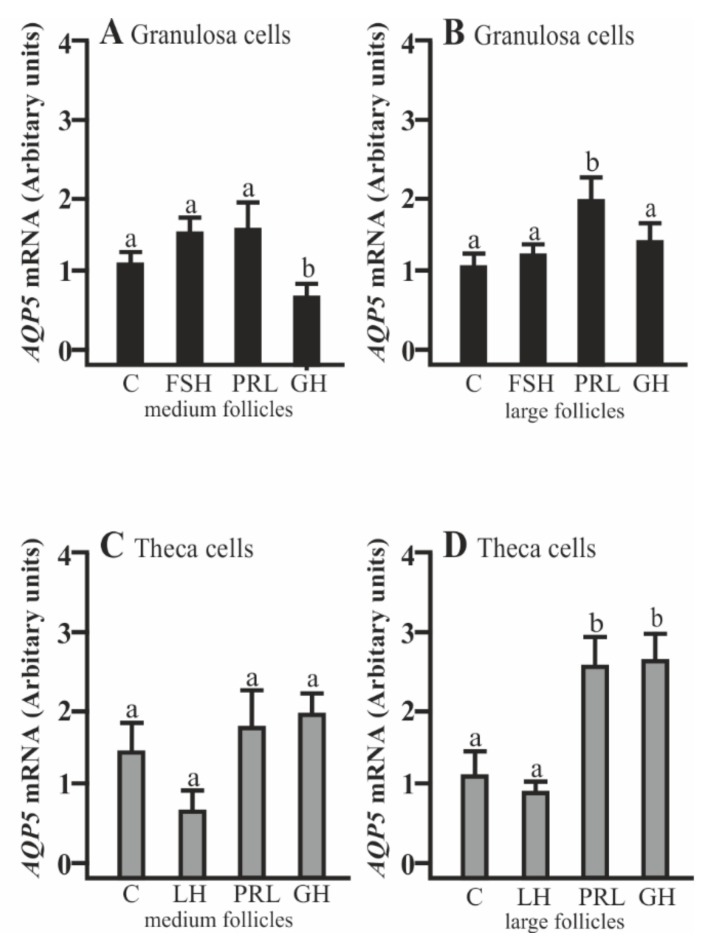
The changes in the expression of Aquaporin 5 (AQP5) mRNA in porcine granulosa (Gc) and theca cells (Tc) obtained from medium (MF) and large (LF) follicles after 24 h incubation in the absence (C) or presence of follicle-stimulating hormone (FSH), luteinizing hormone (LH), prolactin (PRL), and growth hormone (GH). The samples were subjected to real-time qPCR analysis and normalized to the expression of Glyceraldehyde 3-phosphate dehydrogenase (*GAPDH*) and Cyclophylin A (*PPIA*), (**A**,**B**) in the Gc and (**C**,**D**) Tc. Values are expressed as means ± S.E.M from five separate experiments, each performed in triplicate (*p* < 0.05 compared with controls). Statistically significant differences between treatments are indicated by different letters (a, b). C: control.

**Figure 2 ijms-20-04914-f002:**
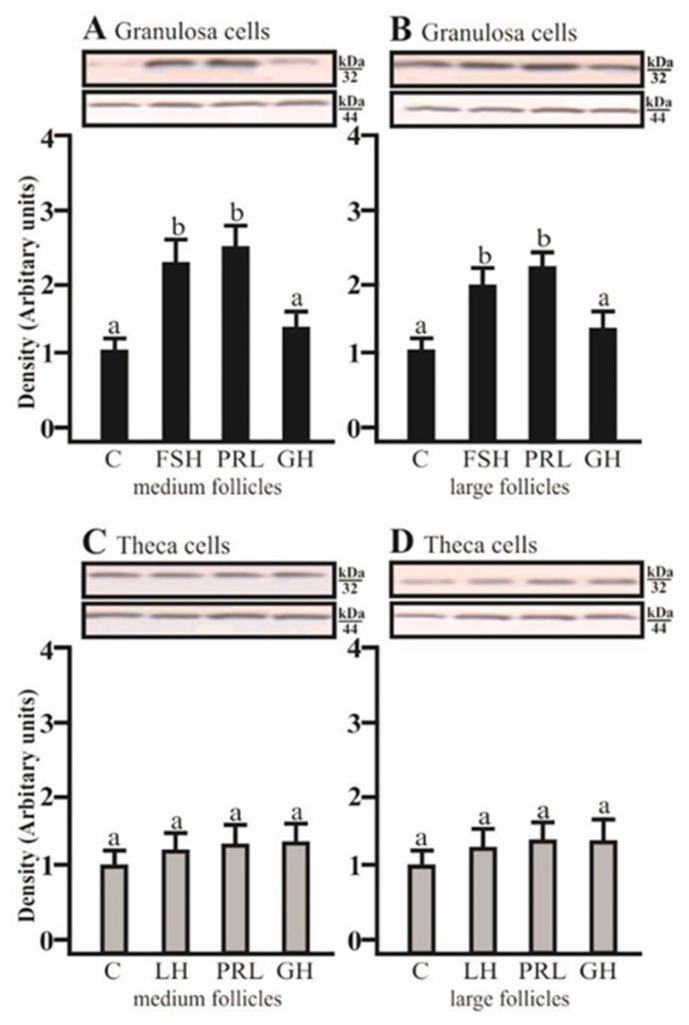
The changes in the expression of AQP5 (kDa 32) protein in porcine Gc and Tc obtained from MF and LF after 24 h incubation in the absence (C) or presence of follicle-stimulating hormone (FSH), luteinizing hormone (LH), prolactin (PRL), and growth hormone (GH). The samples were subjected to Western blot analysis and normalized to the expression of β-actin (kDa 44), (**A**,**B**) in the Gc and (**C**,**D**) Tc. Values are expressed as means ± S.E.M from five separate experiments, each performed in triplicate (*p* < 0.05 compared with controls). Statistically significant differences between treatments are indicated by different letters (a, b). C: control.

**Figure 3 ijms-20-04914-f003:**
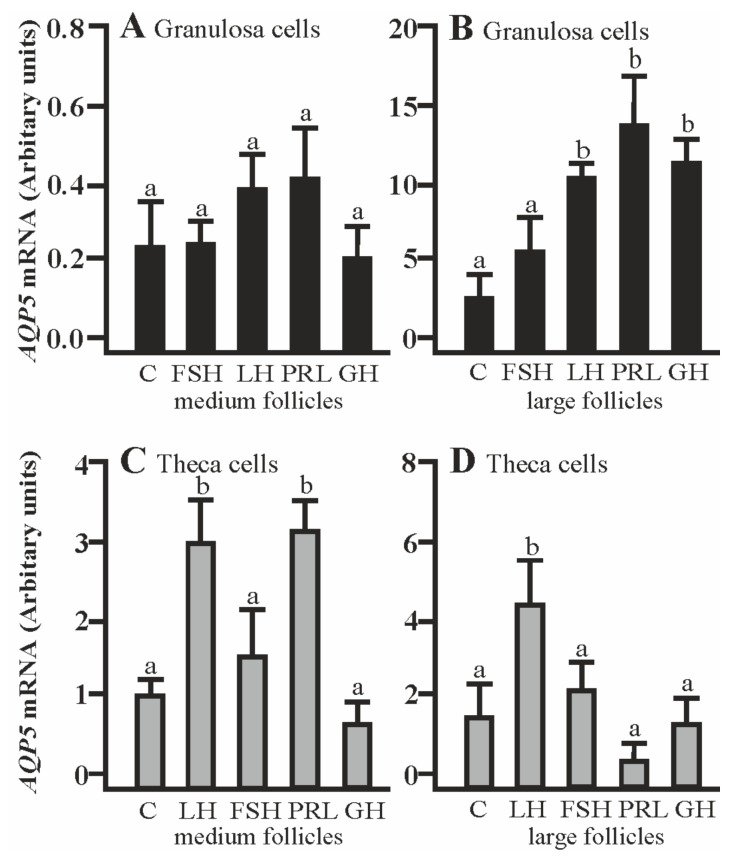
The changes in the expression of AQP5 mRNA in porcine Gc and Tc obtained from MF and LF after 24 h co-culture of these cells in the absence (C) or presence of follicle-stimulating hormone (FSH), luteinizing hormone (LH), prolactin (PRL), and growth hormone (GH). The samples were subjected to real-time qPCR analysis and normalized to the expression of *GAPDH* and *PPIA*, (**A**,**B**) in the Gc and (**C**,**D**) Tc. Values are expressed as means ± S.E.M from five separate experiments, each performed in triplicate (*p* < 0.05 compared with controls). Statistically significant differences between treatments are indicated by different letters (a,b). C: control.

**Figure 4 ijms-20-04914-f004:**
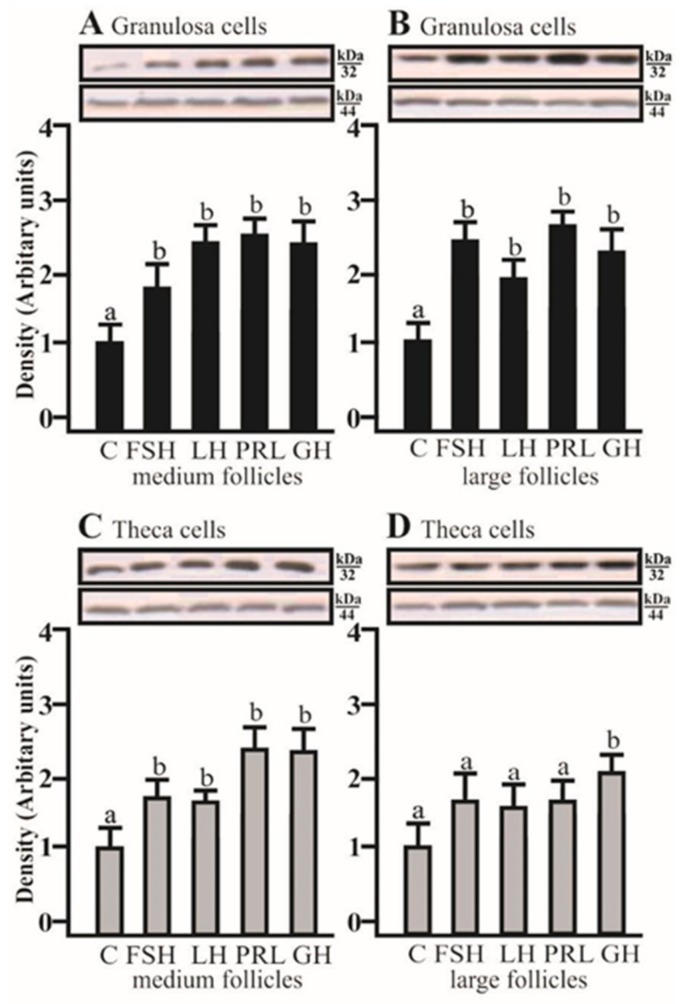
The changes in the expression of AQP5 (kDa 32) protein in porcine Gc and Tc obtained from MF and LF after 24 h co-culture of these cells in the absence (C) or presence of follicle-stimulating hormone (FSH), luteinizing hormone (LH), prolactin (PRL), and growth hormone (GH). The samples were subjected to Western blot analysis and normalized to the expression of β-actin (kDa 44), (**A**,**B**) in the Gc and (**C**,**D**) Tc. Values are expressed as means ± S.E.M from five separate experiments, each performed in triplicate (*p* < 0.05 compared with controls). Statistically significant differences between treatments are indicated by different letters (a, b). C: control.

**Figure 5 ijms-20-04914-f005:**
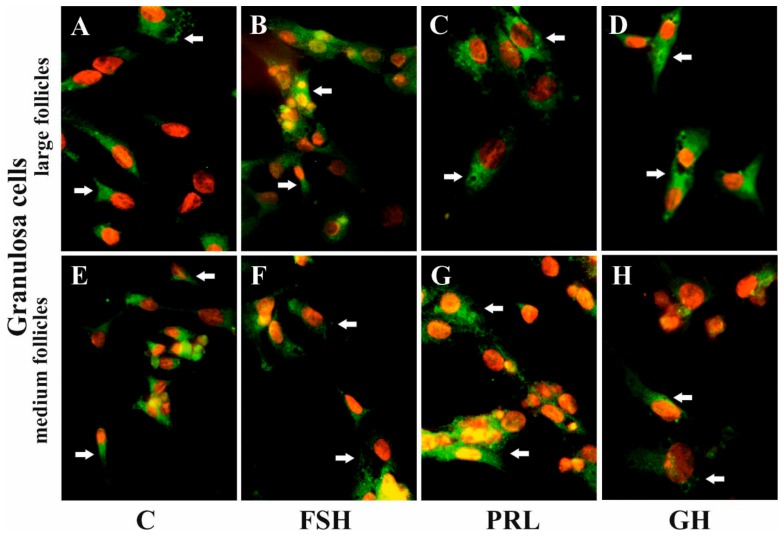
The influence of FSH, PRL, and GH on the subcellular distribution of AQP5 in the Gc of porcine ovarian follicles. After being exposed to FSH (**B,F**), PRL (**C,G**), and GH (**D,H**) for 24 h, Gc were incubated with anti-AQP5 antibody and visualized with Alexa-488 (in green). Nuclei were stained with propidium iodide (in red). Arrow indicates the localization of AQP5 in the cells. The data are representative of five individual experiments, each carried out in duplicate. C: control (**A,E**); magnification of 600×.

**Figure 6 ijms-20-04914-f006:**
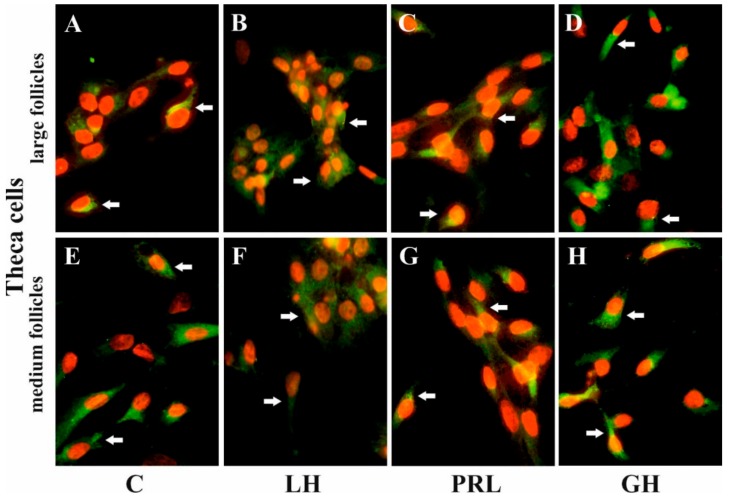
The influence of LH, PRL, and GH on the subcellular distribution of AQP5 in Tc of porcine ovarian follicles. After being exposed to LH (**B,F**), PRL (**C,G**), and GH (**D,H**) for 24 h, Tc were incubated with anti-AQP5 antibody and visualized with Alexa-488 (in green). Nuclei were stained with propidium iodide (in red). Arrow indicates the localization of AQP5 in the cells. The data are representative of five individual experiments, each carried out in duplicate. C: control (**A,E**); magnification of 600×.

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
