# Peer review of "Pituitary Hormones (FSH, LH, PRL, and GH) Differentially Regulate AQP5 Expression in Porcine Ovarian Follicular Cells"

_ijms, 2019, doi:10.3390/ijms20194914_

Round 1

Reviewer 1 Report

I feel it was better than the previous one. However, I feel one problem. It is IF data (Fig. 5). The authors said that a part of AQP5 was also expressed in the plasma membrane. However, it does not seem to be expressed. Identification of subcellular localization is important, because it leads to analogizing the function. I recommend the authors to improve Fig. 5.

Author Response

In response to Referees

The referees are thanked very much for positive and very helpful comments.

Answers to Referee 1 comments

Re.: I feel it was better than the previous one. However, I feel one problem. It is IF data (Fig. 5). The authors said that a part of AQP5 was also expressed in the plasma membrane. However, it does not seem to be expressed. Identification of subcellular localization is important, because it leads to analogizing the function. I recommend the authors to improve Fig. 5.

In my opinion there is a labeling of AQP5 in the cell membranes in the pig ovarian follicular cells. Our previous study (Identification of AQP5 in lipid rafts and its translocation to apical membranes by activation of M3 mAChRs in interlobular ducts of rat parotid gland. Ishikawa Y, Yuan Z, Inoue N, Skowronski MT, Nakae Y, Shono M, Cho G, Yasui M, Agre P, and Nielsen S. American Journal of Physiology Cell Physiology 2005, 289 (5), C1303-11), found by immunoelectron microscopy localization of AQP5 in the cell membranes from control and cevimeline-treated rats.

However, to take account of the reviewer's opinion, we change the description of the results for Figure 5. We changed it from “as well as in the cell membranes” to “a weak labeling of AQP5 was observed in the cell membranes” (in green).

Reviewer 2 Report

The manuscript was modified following exactly the suggestions by the reviewers. Certainly now the manuscript was improved and enriched with new experimental data to reinforce the results obtained and the discussion point. Only a little mistake is still present in the text. The authors should control page 14 of the manuscript because n. 46 of reference list not correspond to a reference so there is a discrepancy between the reference number in the text and in the reference list.

After the changes the manuscript is suitable for publication.

Author Response

In response to Referees

The referees are thanked very much for positive and very helpful comments.

Answers to Referee 2 comments

Re.: The manuscript was modified following exactly the suggestions by the reviewers. Certainly now the manuscript was improved and enriched with new experimental data to reinforce the results obtained and the discussion point. Only a little mistake is still present in the text. The authors should control page 14 of the manuscript because n. 46 of reference list not correspond to a reference so there is a discrepancy between the reference number in the text and in the reference list.

After the changes the manuscript is suitable for publication.

We corrected the number 46 of the references to 51 (in green) in the Materials and Methods. We also corrected numbers 45 and 47 respectively to 50 and 52 (in green; Mat & Met Section).

This manuscript is a resubmission of an earlier submission. The following is a list of the peer review reports and author responses from that submission.

Round 1

Reviewer 1 Report

The manuscript is well-written with some grammatical and typing error. The experiments were well designed, the presentation of the results are clear. The way of the presentation of statistical differences are little bit complicated, but comprehensible. The results are novel linked to the AQP5 regulatation by pituitary hormones in follicular cells.

However, authors must emphasise the significance of their findings in a more obvious way. In the recent form the final conclusion is just a simple description of the results. They have to have a vision about the applicability or importance of these findings, even if it would be partially speculative.

Reviewer 2 Report

The authors examined the effect of FSH, LH, PRL, or GH on the expression of AQP5 mRNA and protein in granulosa and theca cells, both of which were isolated from porcine ovaries. The results suggested a regulatory role of gonadotropins in these cells. The observations are potentially interesting. However, the physiological significance and/or the veterinary clinical relevance of the manuscript is vague. Also, the data appears to be preliminary. For example, the authors did not perform the experiment on subcellular localization of AQP5. Furthermore, the authors failed to present the data concerning the mechanisms by which gonadotropins altered the level of expression of AQP5. Finally, the impact of the manuscript is unfortunately reduced due to a number of grammatical errors and difficulties of phraseology.

Reviewer 3 Report

Dear authors

The argument regarding the possible role of AQPs on different aspects of female reproductive function is of real interest. In the complex, the manuscript is interesting albeit no high data of novelty are introduced. In fact, your recent paper published in 2018 investigate on similar aspects of the present manuscript modifying the studied AQP.  However the manuscript is well written and organized albeit the English form adopted for the manuscript is very basic and require necessarily a complete revision by an English native speaker. Moreover, the authors should add some literature on the theme (es. Zhu et al., Frontiers in Bioscience, Landmark, 20, 838-871, 2015; Huang HF et al., Hum Reprod Update, 2006 12(6):785-95. In addition, regarding the role in “maintaining water homeostasis in reproductive cells” the authors should add a recent paper by Pelagalli et al., 2019 entitled “Cellular distribution of aquaporins in testes of normal and cryptorchid dogs: A preliminary study on dynamic roles”.

Moreover, regarding the figures relative to western blotting the authors should introduce western blot of beta actin in order to control the normalization (Figure 2, Figure 4). Moreover, additional information should be confirmed also by immunohistochemical data.